# The Role of Muscle Density in Predicting the Amputation Risk in Peripheral Arterial Disease: A Tissue Composition Study Using Lower Extremity CT Angiography

**DOI:** 10.3390/diagnostics15111439

**Published:** 2025-06-05

**Authors:** Yueh-Hung Lin, Pei-Shan Tsai, Chung-Lieh Hung, Mirza Faisal Beg, Hung-I Yeh, Chun-Ho Yun, Ming-Ting Wu

**Affiliations:** 1Division of Cardiology, Departments of Internal Medicine, Mackay Memorial Hospital, Taipei 104217, Taiwan; linyehon@gmail.com (Y.-H.L.); jotaro3791@gmail.com (C.-L.H.); hungi.yeh@msa.hinet.net (H.-I.Y.); 2Department of Medicine, Mackay Medical College, New Taipei City 252005, Taiwan; purifying@gmail.com (P.-S.T.); med202657@gmail.com (C.-H.Y.); 3Department of Radiology, Kaohsiung Veterans General Hospital, Kaohsiung 81362, Taiwan; 4Institute of Clinical Medicine, National Yang Ming Chiao Tung University, Taipei 112304, Taiwan; 5Department of Radiology, MacKay Memorial Hospital, Taipei 104217, Taiwan; 6Mackay Junior College of Medicine, Nursing and Management, Taipei City 11260, Taiwan; 7Institute of Biomedical Sciences, Mackay Medical College, New Taipei City 252005, Taiwan; 8School of Engineering Science, Simon Fraser University, Burnaby, BC V5A 1S6, Canada; mfbeg@sfu.ca; 9School of Medicine, National Yang Ming Chiao Tung University, Taipei 112304, Taiwan

**Keywords:** peripheral arterial diseases, computed tomography, artificial intelligence

## Abstract

**Objectives:** Peripheral arterial disease (PAD) is a common vascular condition with amputation as a major complication. While muscle volume and vascular severity is often considered in risk prediction, the prognostic value of muscle density remains underexplored. **Methods**: In this exploratory study, we retrospectively analyzed 134 patients undergoing lower-limb computed tomography angiography between January 2018 and December 2023. Muscle density (MD), muscle volume, intermuscular adipose tissue (IMAT), and vascular severity scores were quantified using deep learning software. Patients were stratified into non-PAD, mild PAD, and critical limb ischemia (CLI) groups. Multivariate Cox regression assessed associations with amputation risk. **Results**: PAD patients, especially those with CLI, had lower muscle volumes (e.g., total leg: 7945.3 ± 2012.5 cm^3^ in CLI vs. 11,161.6 ± 4670.4 cm^3^ in non-PAD), lower muscle densities (e.g., lower leg: 34.0 ± 10.5 HU in CLI vs. 44.1 ± 6.9 HU in non-PAD), higher intermuscular adipose tissue volume percentage (e.g., total leg: 15.6 ± 5.4% in CLI vs. 10.5 ± 3.6% in non-PAD), and higher vascular severity scores (e.g., total leg: 12.6 ± 5.0 in CLI vs. 0.1 ± 0.3 in non-PAD), compared to non-PAD individuals. Only muscle density (MD) including that of abdominal muscle, thigh muscle, and lower leg muscle remained significant predictors of amputation risk after adjusting for confounders. Multivariate Cox regression models, adjusted for demographics and comorbidities, revealed that lower MD of abdomen (<18.7 HU; HR, 6.50, 95% CI, 1.95–21.77), thigh (<27.8 HU; HR, 5.00, 95% CI, 1.60–15.66), and lower leg (<32.4 HU; HR, 6.89, 95% CI, 2.17–21.93) muscles were independently associated with increased amputation risk. **Conclusions**: Muscle density, reflecting muscle quality rather than quantity, was an independent predictor of amputation risk in PAD. These findings highlight the prognostic value of muscle quality and support the integration of advanced imaging techniques, such as automated CTA-based body composition analysis, for risk stratification in PAD patients.

## 1. Introduction

Peripheral arterial disease (PAD) is a manifestation of systemic atherosclerosis, leading to arterial narrowing, reduced blood flow, and ischemic symptoms, which are commonly graded using the Rutherford classification to guide clinical management [1,2,3]. PAD frequently coincides with sarcopenia, characterized by loss of muscle mass and strength [4,5,6].

Lower extremity computed tomography angiography (LECTA) is a widely accepted non-invasive modality for evaluating PAD [7,8,9]. Beyond vascular patency and atherosclerotic burden assessment, LECTA inherently provides information on muscle volume (MV), muscle density (MD), and intermuscular adipose tissue (IMAT), offering valuable insights into tissue composition which might associate with risk of amputation [8,9,10].

Prior studies have used representative slices of LECTA to estimate the MV in PAD [11]. With advancements in deep learning-based automated software [12,13], comprehensive analysis of volumetric body composition (VBC) throughout the lower limbs has become feasible.

Given these developments, this exploratory study aimed to utilize automated software for comprehensive VBC analysis of LECTA to investigate the role of muscle density, alongside other body composition parameters and clinical profiles, in predicting amputation risk in patients with PAD.

## 2. Methods

### 2.1. Study Subjects and Design

This single-center, retrospective exploratory study included patients who underwent contrast-enhanced computed tomography (CT) at MacKay Memorial Hospital, Taipei, Taiwan, between January 2018 and December 2023. Indications for imaging included PAD diagnosis, pre-endovascular therapy evaluation, or cardiovascular assessment in the outpatient setting. PAD was diagnosed according to the 2016 American Heart Association guidelines [6]. Patients with prior vascular interventions, bypass surgery, poor CT image quality, or incomplete clinical data were excluded.

Patients were classified into three groups based on the presence and severity of PAD according to the established PAD criteria and the Rutherford chronic limb ischemia classification [2]: non-PAD, mild PAD (stages I–III), and critical limb ischemia (CLI; stages IV–VI) (Figure 1).

Clinical data and CT images were retrospectively analyzed and de-identified prior to access and analysis by the investigators. The study was approved by the local Institutional Review Board (MacKay Memorial Hospital Institutional Review Board Committee) (24MMHIS116e), and informed consent was waived due to the retrospective nature of the study. The study adhered to the Declaration of Helsinki.

Clinical profiles, including age, sex, height, body weight, and body mass index (BMI), were collected. Each subject’s medical history was thoroughly reviewed for cardiovascular disease (CVD), diabetes mellitus (DM), hypertension, dyslipidemia, chronic kidney disease (CKD), end-stage renal disease (ESRD), and smoking history.

CVD was defined as a documented history of acute coronary syndrome, clinically suspected coronary artery disease, or coronary artery stenosis greater than 50%, as confirmed by coronary CT angiography or coronary angiography. Additionally, cerebrovascular disease, including strokes (cerebrovascular accidents) and transient ischemic attacks, was considered part of CVD.

### 2.2. CT Image Analysis, Processing, and Parameter Measurement

All patients underwent hydration protocols before and after the scan. LECTA was performed using a 256-slice CT scanner (SOMATOM Definition Flash, Siemens, Munich, Germany). CT angiography was performed using a test-bolus tracking method, covering the lower abdominal aorta to the feet. A 15 mL contrast bolus was administered to identify the popliteal artery, with image acquisition initiated 10 s after peak enhancement. Contrast injection included 85 mL of iodinated nonionic contrast medium (Ultravist 300, 300 mg iodine/mL; Bayer Schering, Berlin, Germany), followed by 40 mL of saline at a flow rate of 4 mL/s. Imaging parameters included a collimation of 256 × 0.6 mm, a pitch of 0.9, and tube voltage settings of 100 kVp, with a reference tube current of 220 mA. The acquisition covered the region from the second lumbar vertebra to the feet.

An automated deep learning-based software (Data Analysis Facilitation Suite [DAFS] version 3, Voronoi Health Analytics Inc., Vancouver, Canada) segmented five tissue components from non-contrast CT images between the third lumbar (L3) vertebra to the ankles (Figure 2) [13]. These compositions included skeletal muscle, visceral adipose tissue (VAT), subcutaneous adipose tissue (SAT), IMAT, and bone. A 3rd-year research assistant, trained by an experienced radiologist with over 25 years of experience, manually corrected the automated region of interest if necessary.

Images were segmented from the upper endplate of L3 to the ankle joint and divided into three parts: pelvis (L3 to sacrum), thigh (sacrum to knee), and leg (knee to ankle). Skeletal muscle volume was measured, including abdominal skeletal muscle volume (AMV), bilateral total leg muscle volume (TLMV; thigh + lower leg), bilateral thigh muscle volume (TMV), and bilateral lower leg muscle volume (LLMV). These segmentations were based on deep learning-based software with a range from −29 to +150 Hounsfield units (HU), as previously validated [11,14].

Additionally, abdominal subcutaneous adipose tissue volume (SATV) and abdominal visceral fat volume (VATV) were measured using deep learning-based contouring, with a range of −190 to −30 HU. IMAT was defined as fat located between muscles but beneath the fascia. IMAT volumes were calculated for the abdomen [IMATV(ABD)], bilateral total leg [IMATV(TL)], bilateral thigh [IMATV(TM)], and bilateral lower leg [IMATV(LL)]. IMATV percentage was determined by dividing IMATV by the sum of muscle volume and IMATV, with values calculated for the abdomen, total leg, thigh, and lower leg. The mean CT attenuation of MD was also measured for the abdomen, thigh, and lower leg. To account for body size, muscle volume index (muscle volume/height^2^ [m^2^]) and fat volume index (fat volume/height^2^ [m^2^]) were also calculated [11,15].

CTA severity was assessed by scoring arterial stenosis across the iliac, upper leg (femoropopliteal) arteries, and lower leg arteries (anterior tibial, peroneal, and posterior tibial arteries) [11]. Stenosis severity was graded as 0 (no stenosis), 1 (<50% stenosis), 2 (≥50% stenosis), and 3 (total occlusion). Total scores were summed, with a possible range of 0 to 24.

### 2.3. Laboratory Measurements

Overnight fasting blood samples were analyzed for glucose levels and renal function. The estimated glomerular filtration rate (eGFR) was calculated using the Modification of Diet in Renal Disease (MDRD) equation.

### 2.4. Outcome Measure

The primary clinical outcome was the occurrence of any amputation (minor or major) during follow-up after the CT scan.

### 2.5. Statistical Analysis

Continuous variables were expressed as mean ± standard deviation (SD) and compared using an analysis of variance (ANOVA). Categorical variables were presented as counts (*n*) and percentages (%) and compared using the chi-square test. Non-normally distributed data were analyzed with the Mann–Whitney U test.

The association between MV, MD, IMATV, amputation events, and PAD was evaluated using univariate and multivariate logistic regression, adjusting for potential confounders, including age, sex, height, hypertension, DM, dyslipidemia, CVD, CKD, and smoking history.

Receiver operating characteristic (ROC) curve analysis was performed to evaluate amputation events, with the optimal cut-off value determined using the area under the ROC curve (AUC) to optimal sensitivity and specificity. Cox regression models were used to analyze survival outcomes, and Kaplan–Meier survival curves were generated to compare risk groups based on the ROC curve cut-off value. All statistical analyses were two-tailed, and *p* < 0.05 was considered statistically significant. Statistical analyses were performed using IBM SPSS Statistics for Mac, version 26.0 (IBM Corp., Armonk, NY, USA, released in 2019).

## 3. Results

### 3.1. Baseline Characteristics

During the study period, 178 patients were initially enrolled. Of these, 44 were excluded due to poor image quality that prevented successful reconstruction using the deep learning-based software. The remaining 134 patients included 33 patients without PAD, 48 with mild PAD, and 53 with CLI PAD. Among the CLI PAD patients, 17 underwent amputation. The median follow-up duration after CT imaging was 3.24 years.

Table 1 provides an overview of the baseline clinical characteristics of the study population. Compared to the non-PAD patients, those with mild PAD and CLI PAD were older and had lower body weight. Additionally, the prevalence of CVD, DM, atrial fibrillation, CKD, and ESRD was higher in the mild PAD and CLI PAD groups than in the non-PAD group. The patients in these groups also had lower eGFR than those without PAD.

### 3.2. Associations Between Muscle Volume, Muscle Density, and PAD

AMV and PMV were strongly correlated with TLMV (r = 0.835 and 0.870, respectively; both *p* < 0.001) and TMV (r = 0.838 and 0.880, respectively; both *p* < 0.001). A strong positive correlation was also observed between AMV, PMV, and LLMV (r = 0.756 and 0.765, respectively; both *p* < 0.001).

Figure 3A illustrates the association between PAD and MV, indicating that the patients with PAD had significantly lower TLMV, TMV, and LLMV compared to the non-PAD patients (all *p* < 0.001).

Compared to the non-PAD patients, those with mild PAD and CLI PAD had significantly lower values for AMV, AMV/Height^2^, PMV, PMV/Height^2^, TLMV, TLMV/Height^2^, TMV, TMV/Height^2^, LLMV, and LLMV/Height^2^ (all *p* < 0.05) (Table 2). Furthermore, the patients with CLI PAD had significantly lower TLMV, LLMV, and LLMV/Height^2^ than those with mild PAD (all *p* < 0.001). However, there were no significant differences between the mild PAD and CLI PAD groups in AMV, AMV/Height^2^, PMV, PMV/Height^2^, TLMV/Height^2^, TMV, or TMV/Height^2^ (Table 2).

Figure 3B shows that the PAD patients had significantly lower MD in the abdomen, thigh, and lower leg compared to the non-PAD patients (all *p* < 0.001). The CLI PAD group had significantly lower MD in these regions than the non-PAD group (*p* < 0.05). Additionally, the CLI PAD patients had significantly lower thigh and lower leg MD than those with mild PAD (all *p* < 0.05). The patients with mild PAD also demonstrated lower thigh MD compared to the non-PAD individuals (*p* < 0.05) (Table 2).

### 3.3. Associations Between IMATV, IMATV Percentage, and PAD

A strong positive correlation was observed between IMATV(ABD) and IMATV(TM) (r = 0.715) and a moderate correlation between IMATV(ABD) and IMATV(LL) (r = 0.416) (both *p* < 0.001).

When comparing the PAD and non-PAD patients, no significant differences were found in IMATV, SATV, or VATV across the PAD severity groups, including those with mild PAD and CLI PAD. However, the PAD patients had significantly higher IMAT percentages, including IMATV(TL)%, IMATV(TM)%, and IMATV(LL)%, compared to the non-PAD patients (all *p* < 0.001) (Figure 3C).

Furthermore, the CLI PAD group demonstrated significantly higher IMATV(TL)%, IMATV(TM)%, and IMATV(LL)% compared to the non-PAD group (all *p* < 0.05), while no significant differences were observed between the non-PAD and mild PAD groups. IMATV(ABD)% did not significantly differ among the three groups (Table 2).

### 3.4. Association Between PAD and Amputation Events

Table 3 presents the results of univariate and multivariate logistic regression analyses for amputation prediction in the PAD patients. In univariate analysis, several factors were significantly associated with a higher amputation risk, including increased IMATV percentage in the total leg (OR, 1.11, 95% CI, 1.01–1.22) and thigh (OR, 1.10, 95% CI; 1.01–1.19), as well as greater vascular stenosis severity in the total leg (OR, 1.11, 95% CI, 1.02–1.21) and lower leg (OR, 1.14, 95% CI, 1.02–1.28). Lower muscle density also emerged as a significant predictor, with associations observed for abdomen MD (OR, 0.89, 95% CI, 0.83–0.96), thigh MD (OR, 0.76, 95% CI, 0.66–0.89), and lower leg MD (OR, 0.93, 95% CI, 0.88–0.98). However, muscle volume and iliac and upper leg artery stenosis severity as well as other fat volume parameters were not significantly associated with amputation risk.

The multivariate logistic regression models were adjusted for potential confounders: Model 1 (age and sex), Model 2 (age, sex, and height), and Model 3 (age, sex, height, hypertension, DM, dyslipidemia, CVD, CKD, and smoking history). After adjustments, increased IMATV percentage and vascular stenosis severity were no longer significant predictors of amputation. However, muscle density remained an independent predictor, with abdomen MD (OR, 0.85, 95% CI, 0.76–0.95), thigh MD (OR, 0.77, 95% CI, 0.64–0.93), and lower leg MD (OR, 0.92, 95% CI, 0.86–0.98) retaining their significance.

Figure 4 shows the ROC curve analysis, which identified optimal thresholds for predicting amputation at 18.7 HU for abdomen MD, 27.8 HU for thigh MD, and 32.4 HU for lower leg MD. For abdomen MD, the area under the curve (AUC) was 0.739 (*p* = 0.002), with a sensitivity of 47.1%, specificity of 85.7%, a positive predictive value (PPV) of 40.4%, and a negative predictive value (NPV) of 88.9%. Thigh MD showed an AUC of 0.763 (*p* = 0.001), with a sensitivity of 58.8%, specificity of 84.5%, PPV of 43.5%, and NPV of 91.0%. Lower leg MD showed an AUC of 0.751 (*p* = 0.001), with a sensitivity of 64.7%, specificity of 79.8%, PPV of 39.2%, and NPV of 91.7%.

Figure 5 shows Kaplan–Meier survival curves, demonstrating a significantly higher cumulative incidence of amputation in patients with MD below these thresholds. The patients with abdomen MD < 18.7 HU (log-rank *p* = 0.001), thigh MD < 27.8 HU (log-rank *p* < 0.001), and lower leg MD < 32.4 HU (log-rank *p* < 0.001) had significantly higher amputation rates.

In multivariate Cox regression models adjusted for demographics (age, sex, height) and comorbidities (CVD, hypertension, DM, dyslipidemia, CKD, smoking history), lower MD remained an independent predictor of amputation risk. Specifically, abdomen MD < 18.7 HU (HR, 6.50, 95% CI, 1.95–21.77), thigh MD < 27.8 HU (HR, 5.00, 95% CI, 1.60–15.66), and lower leg MD < 32.4 HU (HR, 6.89, 95% CI, 2.17–21.93) were significant predictors of amputation risk (Figure 5).

## 4. Discussion

This study presents the first comprehensive volumetric quantitative CT analysis of tissue composition, including MV, MD, IMATV, and vascular severity, from L3 to the ankles, aimed at predicting the amputation outcomes in patients with PAD. Key findings include the following: (1) The CLI patients showed significantly lower-leg muscle atrophy compared to the mild PAD patients. However, abdominal and thigh MV, as well as MV index, showed no differences between the mild PAD and CLI groups. (2) The PAD patients, especially those with CLI, had higher IMATV percentages and reduced MD in the lower leg and thigh than the non-PAD patients. (3) Reduced MD, rather than MV, IMATV, or vascular stenosis severity was an independent predictor of amputation risk. These findings suggest MD, reflecting muscle quality via intramuscular fat infiltration, may serve as a novel imaging biomarker for assessing amputation risk in PAD patients.

PAD is characterized by atherosclerotic narrowing of the lower extremity arteries, leading to impaired blood flow and progressive muscle pathology. This includes reduced muscle area, increased fatty infiltration, fibrosis, and metabolic abnormalities, collectively resulting in muscle dysfunction and sarcopenia [16]. Sarcopenia, in turn, is associated with reduced quality of life, as well as an increased risk of limb amputation and mortality [17,18,19]. Previous research has established the importance of muscle composition in PAD but has focused primarily on muscle volume without considering MD or IMAT. For instance, Tsai et al. [11] used CT imaging to assess muscle and fat composition in different anatomical regions, showing that muscle atrophy occurs earlier in the lower extremities of PAD patients, especially those with severe arterial stenosis. However, they did not investigate MD or IMAT and the association with outcome. Similarly, Matsubara et al. [20] found that sarcopenia, defined by reduced muscle area at the L3 vertebra, predicted poor survival in patients with CLI; however, MD and IMAT were not assessed.

We observed significant muscle loss in the lower legs of PAD patients, particularly in those with CLI, while the proximal thigh and abdominal muscles remained relatively preserved. This pattern reflects the direct impact of reduced blood flow and arterial stenosis on distal muscles. These findings align with previous research [11] showing that peripheral muscle loss is more pronounced in PAD than central atrophy, underscoring the disease’s localized effects.

It is of note that we used a non-contrast CT scan, not a CTA scan of LECTA, for muscle density analysis [21,22]. Thus, we can avoid the mixture effect on MD resulting from perfusion difference in CTA, which is dependent on the patients’ vascularity and contrast injection protocol.

In addition, we separated the measurement of intramuscular fat (MD) from intermuscular fat (IMAT) in this study. This could be accomplished only by an advanced deep learning algorithm from the software. On the contrary, many studies mixed these two entities as one measurement as documented in a systemic review [23]. We found that while total IMAT volume was similar between the PAD and non-PAD patients, the percentage of IMAT to MV was significantly higher in the PAD patients, particularly those with CLI. This result supported the appropriation of separating IMAT and MD in the analysis.

Reduced MD, often driven by increased intramuscular fat, or called myosteatosis [24,25], is associated with impaired mobility, diminished strength, and higher mortality. For example, Vedder et al. [26] found that myosteatosis, assessed via MD at the L3 vertebra, was an independent predictor of survival in PAD patients. Similarly, Kirsten et al. [9] identified MD thresholds in the lower extremities as predictors of amputation-free survival. Together with our finding, it could be concluded that muscle quality, rather than quantity, plays a critical role in determining clinical outcomes in PAD [9,23,26,27].

Unlike studies using only one to three representative CT slices [11], our volumetric approach, analyzing entire muscle compartments from L3 to the ankles, avoided sampling bias and allowed for the generation of comprehensive 3D muscle maps to better evaluate the regional impact of PAD.

The clinical impact of reduced MD has been supported by the laboratory study of microscopic myosteatosis. The metabolic and physiological changes associated with PAD contribute to an increase in insulin resistance, pro-inflammatory cytokines, chronic inflammation, and mitochondrial dysfunction [28,29] and drive the accumulation of lipid intermediates, such as diacylglycerol and ceramide [30,31,32]. These mechanisms are also observed in sarcopenia, where disrupted anabolic signaling and increased catabolic activity drive progressive muscle loss [23]. This cycle of metabolic dysfunction underpins the development of myosteatosis and manifests as reduced MD grossly on CT in PAD.

However, the causal effect of PAD and reduced MD in our subjects is still not explicitly identified. It could be directly ischemia-induced metabolic changes as described above, or it could be indirectly due to ischemia-related physical constraint. Lastly, there may be a common systemic risk factors that causes PAD vasculopathy and MD reduction simultaneously.

### Limitations

There were several limitations to this study. First, as a retrospective, cross-sectional analysis, the study cannot establish a causal relationship between PAD, reductions in MV and MD, and clinical outcomes. Second, patients who had previously undergone endovascular therapy or surgical bypass were excluded, potentially introducing selection bias and limiting the generalizability of the findings to all PAD patients, including those who had received revascularization. Third, the relatively small sample size may limit the statistical power and external validity of the study. However, a post hoc power analysis based on amputation outcomes yielded a power of 99.7%, with a minimum required sample size of 19 patients per group (total *n* = 38), indicating adequate statistical strength to support the primary findings. Fourth, the objective physical-activity data (e.g., step counters or formal questionnaires) were not available in the registry, and reduced ambulation may have contributed to lower MD. However, MD remained an independent predictor even after adjusting for mobility-limiting comorbidities, suggesting an ischemia-driven effect beyond deconditioning. Lastly, an independent validation cohort was not included, which may limit the generalizability of our findings. Future prospective studies with larger patient cohorts are needed to validate these findings and further explore the prognostic value of MD in PAD.

## 5. Conclusions

This study highlights the significant role of MD as a biomarker for PAD severity and amputation risk. By emphasizing muscle quality over quantity and integrating advanced imaging techniques, our findings provide a more comprehensive understanding of muscle composition in PAD. These insights reinforce the prognostic value of MD and underscore its potential to guide personalized care strategies aimed at improving outcomes in this patient population.

## Figures and Tables

**Figure 1 diagnostics-15-01439-f001:**
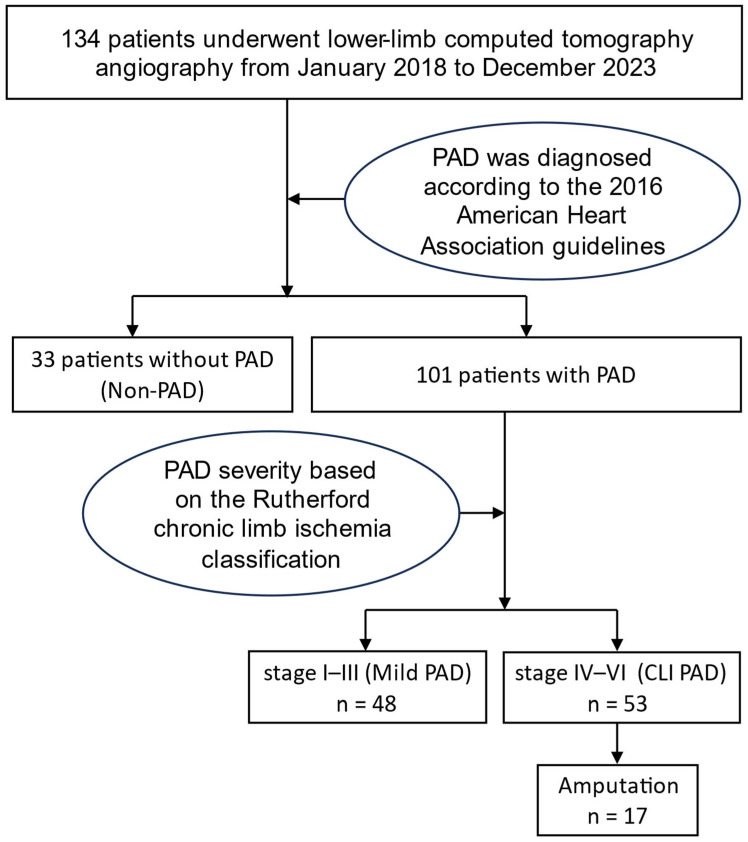
Flowchart illustrating patient selection. Abbreviations: PAD, peripheral arterial disease; CLI, critical limb ischemia.

**Figure 2 diagnostics-15-01439-f002:**
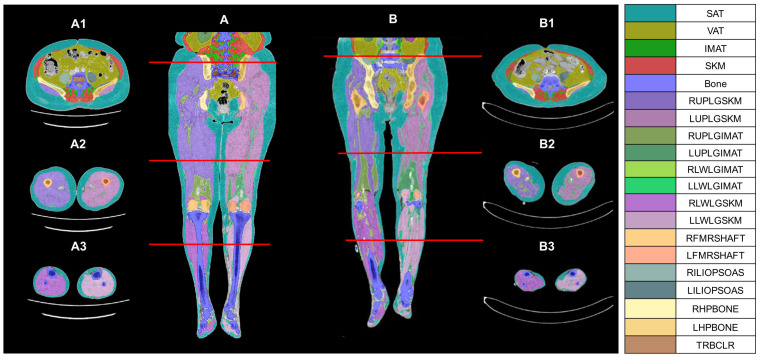
Two representative cases of color CT volumetric body composition maps. (**Panel A**): an 84-year-old woman, BMI (27.2 kg/m^2^) without PAD. (**Panel B**): an 88-year-old woman (BMI 25.2 kg/m^2^) with severe PAD. (**A**,**B**): coronary views from L3 to the ankles. (**A1**–**A3**,**B1**–**B3**): the corresponding cross-sectional views at the three red cutlines, respectively. Color tissue composition maps clearly show the differences between (**A**,**B**), such as increased intermuscular fat infiltration and reduced skeletal muscle volume and density, particularly in the lower extremities in (**Panel B**). Abbreviations: PAD, peripheral arterial disease; CT, computed tomography; SAT, subcutaneous adipose tissue; VAT, visceral adipose tissue; IMAT, intramuscular adipose tissue; SKM, skeletal muscle; RUPLGSKM, right upper leg skeletal muscle; LUPLGSKM, left upper leg skeletal muscle; RUPLGIMAT, right upper leg IMAT; LUPLGIMAT, left lower leg IMAT; RLWLGIMAT, right lower leg IMAT; LLWLGIMAT, left lower leg IMAT; RLWLGSKM, right lower leg skeletal muscle; LLWLGSKM, left lower leg skeletal muscle; RFMRSHAFT, right femoral shaft; LFMRSHAFT, left femoral shaft; RILIOPSOAS, right iliopsoas; LILIOPSOAS, left iliopsoas; RHPBONE, right hip bone; LHPBONE, left hip bone; TRBCLR, trabecular bone.

**Figure 3 diagnostics-15-01439-f003:**
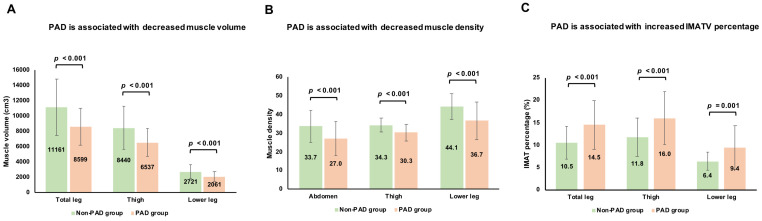
Associations of PAD with muscle volume, IMATV percentage, and muscle density. (**A**) Association between PAD and muscle volume, including total leg, thigh, and lower leg. (**B**) Association between PAD and muscle density, including abdomen, thigh, and lower leg. (**C**) Association between PAD and IMATV percentage, including total leg, thigh, and lower leg. Abbreviations: PAD, peripheral arterial disease; IMATV, intermuscular adipose tissue volume.

**Figure 4 diagnostics-15-01439-f004:**
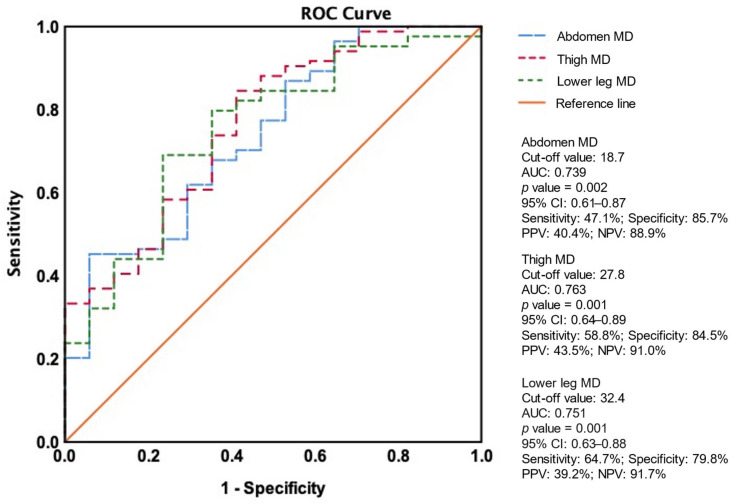
ROC curves for the prediction of amputation events. Abbreviations: ROC, receiver operating characteristic curve; AUC, area under the receiver operating characteristic curves; PPV, positive predictive value; NPV, negative predictive value; MD, muscle density.

**Figure 5 diagnostics-15-01439-f005:**
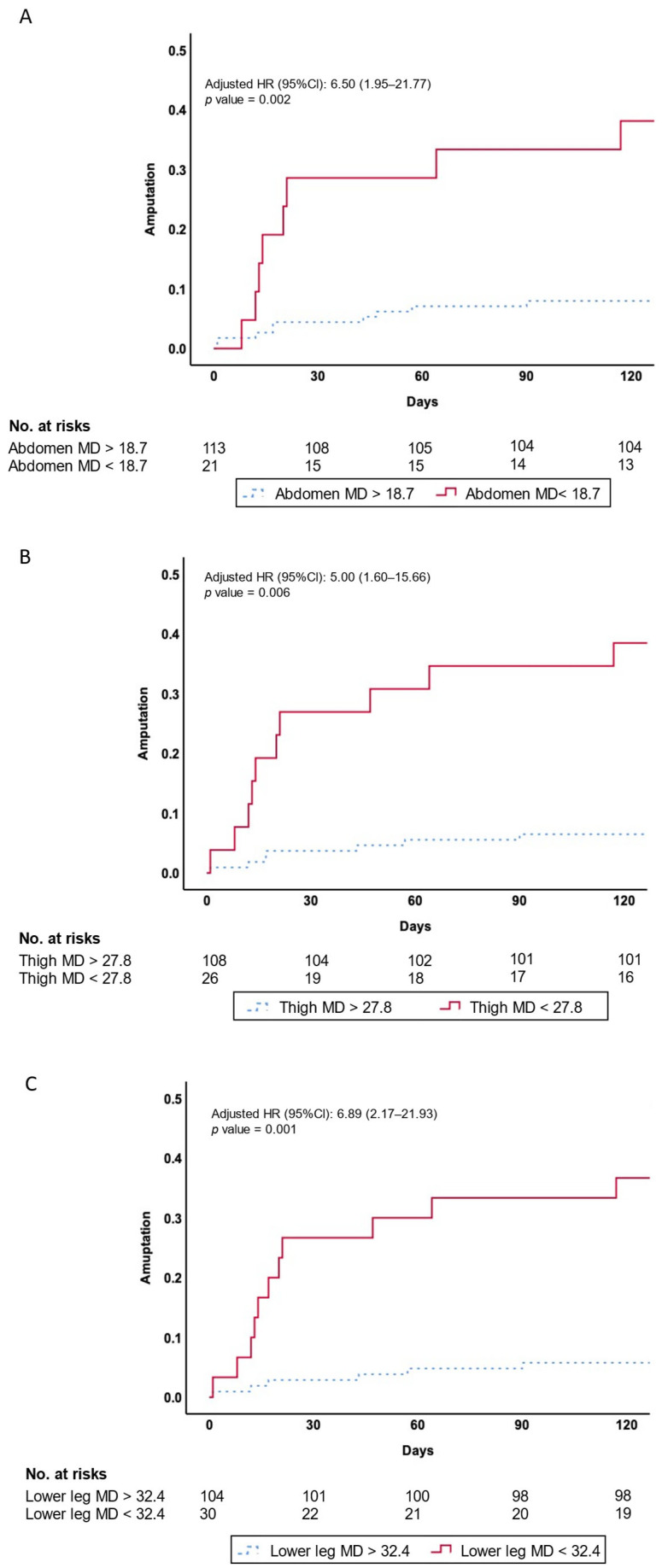
Kaplan–Meier plot of amputation. Kaplan–Meier survival curves showing amputation events according to (**A**), thigh (**B**), and lower leg (**C**) MD in the present study. Abbreviations: MD, muscle density; HR, hazard rate.

**Table 1 diagnostics-15-01439-t001:** Patients’ characteristics.

	Non-PAD	Mild PAD	CLI PAD	*p* Value
Number of patients (*n*)	33	48	53	
Age (years)	66.4 ± 12.8	69.8 ± 10.3	73.1 ± 10.71	0.026
Sex (male, %)	19 (57.6%)	34 (70.8%)	31 (58.5%)	0.345
Height (cm)	159.8 ± 9.0	162.6 ± 8.4	159.1 ± 9.2	0.121
Weight (kg)	70.2 ± 14.7	64.2 ± 11.1	61.7 ± 12.3	0.010
BMI	27.5 ± 5.2	24.3 ± 3.7	24.3 ± 4.0	0.001
Medical history				
CVD (%)	5 (15.2%)	22 (45.8%)	12 (22.6%)	0.005
Hypertension (%)	25 (75.8%)	43 (89.6%)	46 (86.8%)	0.207
DM (%)	14 (42.4%)	25 (52.1%)	43 (81.8%)	<0.001
Dyslipidemia (%)	22 (66.7%)	28 (58.3%)	26 (49.1%)	0.266
Atrial fibrillation (%)	2 (6.3%)	7 (17.1%)	16 (33.3%)	0.011
CKD (%)	8 (24.2%)	28(58.3%)	37(69.8%)	<0.001
ESRD (%)	3 (10.3%)	7 (24.1%)	19 (65.5%)	0.005
Smoking history (%)	13 (39.4%)	22(45.8%)	21 (39.6%)	0.893
Laboratory data				
HbA1C (%)	6.4 ± 2.0	7.2 ± 1.9	7.3 ± 1.5	0.063
eGFR (MDRD), (mL/min/1.73 m^2^)	70.6 ± 29.2	58.3 ± 28.4	43.1 ± 32.6	<0.001

PAD, peripheral arterial disease; CLI, critical limb ischemia; BMI, body mass index; CVD, cardiovascular disease; DM, diabetes melilites; CKD, chronic kidney disease; ESRD, end-stage renal disease; HbA1C, glycated hemoglobin; eGFR, estimated glomerular filtration rate.

**Table 2 diagnostics-15-01439-t002:** CT parameters of the three patient groups.

	Non-PAD Group	Mild PAD	CLI PAD	*p* Value
Number of patients (*n*)	33	48	53	
Muscle volume				
AMV (cm^3^)	1120.8 ± 408.8	943.29 ± 252.7 *	920.6 ± 261.0 ^@^	0.008
AMV/Ht^2^ (cm^3^/m^2^)	432.6 ± 132.4	355.2 ± 84.2 *	361.0 ± 87.4 ^@^	0.001
PMV (cm^3^)	620.0 ± 231.4	528.3 ± 152.0 *	502.3 ± 26.5 ^@^	0.006
PMV/Ht^2^ (cm^3^/m^2^)	238.1 ± 72.7	197.6 ± 47.1 *	196.2 ± 36.3 ^@^	<0.001
TLMV (cm^3^)	11,161.6 ± 4670.4	9320.9 ± 2601.2 *	7945.3 ± 2012.5 ^@#^	<0.001
TLMV/Ht^2^ (cm^3^/m^2^)	4314.9 ± 1164.4	3502.5 ± 846.1 *	3121.3 ± 659.4 ^@^	<0.001
TMV (cm^3^)	8440.4 ± 2833.9	7040.6 ± 2037.4 *	6082.2 ± 1482.4 ^@^	<0.001
TMV/Ht^2^ (cm^3^/m^2^)	3259.9 ± 900.2	2644.7 ± 664.3 *	2390.9 ± 491.3 ^@^	<0.001
LLMV (cm^3^)	2721.2 ± 877.4	2280.3 ± 626.4 *	1863.1 ± 581.4 ^@#^	<0.001
LLMV/Ht^2^ (cm^3^/m^2^)	1055.0 ± 284.2	857.7 ± 208.6 ^@^	730.4 ± 191.6 ^@#^	<0.001
Muscle density				
Abdomen MD (HU)	33.7 ± 8.6	29.1 ± 9.4	25.1 ± 8.4 ^@^	<0.001
Thigh MD (HU)	34.3 ± 3.8	31.9 ± 4.3 *	28.8 ± 4.2 ^@#^	<0.001
Lower leg MD (HU)	44.1 ± 6.9	39.6 ± 8.5	34.0 ± 10.5 ^@#^	<0.001
Fat volume				
SATV (cm^3^)	4062.1 ± 1822.9	3426.8 ± 1625.4	3186.1 ± 1647.1	0.064
SATV/Ht^2^ (cm^3^/m^2^)	1627.5 ± 807.6	1311.9 ± 635.0	1281.8 ± 674.4	0.062
VATV (cm^3^)	2075.9 ± 1004.8	1886.9 ± 886.0	1949.5 ± 1180.9	0.722
VATV/Ht^2^ (cm^3^/m^2^)	827.5 ± 440.2	716.4 ± 336.3	778.5 ± 494.3	0.510
IMATV(ABD) (cm^3^)	163.3 ± 45.8	159.4 ± 53.3	161.8 ± 5.7	0.935
IMATV(ABD)/Ht^2^ (cm^3^/m^2^)	64.8 ± 20.8	60.8 ± 20.7	64.4 ± 18.9	0.571
IMATV(TL) (cm^3^)	1232.6 ± 366.4	1355.1 ± 480.8	1428.6 ± 497.8	0.165
IMATV(TL)/Ht^2^ (cm^3^/m^2^)	486.2 ± 155.6	516.5 ± 189.6	567.5 ± 197.4	0.122
IMATV(TM) (cm^3^)	1053.6 ± 316.7	1157.1 ± 420.9	1228.9 ± 427.5	0.147
IMATV(TM)/Ht^2^ (cm^3^/m^2^)	416.6 ± 138.2	441.6 ± 167.8	489.1 ± 172.4	0.111
IMATV(LL) (cm^3^)	179.0 ± 63.5	198.0 ± 76.9	199.7 ± 100.2	0.498
IMATV(LL)/Ht^2^ (cm^3^/m^2^)	69.6 ± 22.8	74.8 ± 28.5	78.4 ± 36.4	0.437
IMATV percentage				
IMATV(ABD) (%)	13.9 ± 5.2	15.1 ± 5.7	15.5 ± 4.7	0.357
IMATV(TL) (%)	10.5 ± 3.6	13.2 ± 5.2	15.6 ± 5.4 ^@^	<0.001
IMATV(TM) (%)	11.8 ± 4.3	14.7 ± 5.9	17.1 ± 5.7 ^@^	<0.001
IMATV(LL) (%)	6.4 ± 2.0	8.5 ± 4.0	10.2 ± 5.5 ^@^	0.001
Vascular severity score				
Vascular severity score (iliac)	0.2 ± 0.5	13.5 ± 7.8 *	18.4 ± 7.0 ^@^	<0.001
Vascular severity score (TL)	0.1 ± 0.3	8.8 ± 5.8 *	12.6 ± 5.0 ^@#^	<0.001
Vascular severity score (UL)	0.1 ± 0.3	2.0 ± 1.9 *	1.5 ± 1.6 ^@^	<0.001
Vascular severity score (LL)	0.0 ± 0.2	4.7 ± 2.7 *	5.8 ± 3.2 ^@#^	<0.001

* *p* < 0.05, comparison between non-PAD and mild PAD groups; @ *p* < 0.05, comparison between non-PAD and CLI PAD groups; # *p* < 0.05, comparison between mild PAD and CLI PAD groups. PAD, peripheral arterial disease; CLI, critical limb ischemia; AMV, abdominal skeletal muscle volume; Ht, height; PMV, psoas muscle volume; TLMV, total leg muscle volume; TMV, thigh muscle volume; LLMV, lower leg muscle volume; MD, muscle density; VATV, visceral adipose tissue volume; SATV, subcutaneous adipose tissue volume; IMATV, intermuscular adipose tissue volume; IMATV(ABD), abdominal intermuscular adipose tissue volume; IMATV(TL), total leg intermuscular adipose tissue volume; IMATV(TM), thigh intermuscular adipose tissue volume; IMATV(LL), lower leg intermuscular adipose tissue volume; TL, total leg; UL, upper leg; LL, lower leg.

**Table 3 diagnostics-15-01439-t003:** Prediction of amputation in PAD groups.

	Unadjusted		Model 1		Model 2		Model 3	
	OR, 95% CI	*p* Value	OR, 95% CI	*p* Value	OR, 95% CI	*p* Value	OR, 95% CI	*p* Value
AMA	1.00 (1.00–1.00)	0.851	1.00 (1.00–1.00)	0.333	1.00 (1.00–1.00)	0.471	1.00 (1.00–1.01)	0.570
PMV	1.00 (0.99–1.00)	0.286	1.00 (1.00–1.00)	0.747	1.00 (1.00–1.01)	0.877	1.00 (0.99–1.01)	0.877
TLMV	1.00 (1.00–1.00)	0.189	1.00 (1.00–1.00)	0.526	1.00 (1.00–1.00)	0.298	1.00 (1.00–1.00)	0.540
TMV	1.00 (1.00–1.00)	0.263	1.00 (1.00–1.00)	0.737	1.00 (1.00–1.00)	0.497	1.00 (1.00–1.00)	0.828
LMV	1.00 (1.00–1.00)	0.080	1.00 (1.00–1.00)	0.181	1.00 (1.00–1.00)	0.062	1.00 (1.00–1.00)	0.117
Abdomen MD	0.89 (0.83–0.96)	0.001	0.87 (0.79–0.96)	0.004	0.86 (0.78–0.95)	0.002	0.85 (0.76–0.95)	0.003
Thigh MD	0.76 (0.66–0.89)	<0.001	0.78 (0.67–0.92)	0.004	0.77 (0.65–0.92)	0.004	0.77 (0.64–0.93)	0.007
Lower leg MD	0.93 (0.88–0.98)	0.005	0.92 (0.87–0.98)	0.005	0.92 (0.87–0.98)	0.006	0.92 (0.86–0.98)	0.007
IMATV(ABD) (%)	1.07 (0.97–1.17)	0.189	1.07 (0.94–1.21)	0.339	1.07 (0.94–1.21)	0.324	1.10 (0.95–1.27)	0.198
IMATV(TL) (%)	1.11 (1.01–1.22)	0.026	1.09 (0.98–1.21)	0.118	1.11 (0.99–1.24)	0.080	1.08 (0.95–1.22)	0.244
IMATV(TM) (%)	1.10 (1.01–1.19)	0.037	1.07 (0.97–1.18)	0.192	1.08 (0.98–1.20)	0.138	1.05 (0.94–1.18)	0.385
IMATV(LL) (%)	1.09 (0.99–1.20)	0.077	1.10 (1.00–1.21)	0.061	1.10 (1.00–1.22)	0.053	1.11 (0.99–1.24)	0.080
Vascular severity score (iliac)	0.95 (0.70–1.29)	0.747	1.06 (0.77–1.45)	0.927	0.99 (0.72–1.38)	0.971	1.07 (0.74–1.55)	0.732
Vascular severity score (TL)	1.11 (1.02–1.21)	0.019	1.10 (1.01–1.20)	0.032	1.11 (1.02–1.22)	0.022	1.09 (0.99–1.21)	0.078
Vascular severity score (UL)	1.20 (0.99–1.46)	0.059	1.21 (0.99–1.47)	0.060	1.22 (1.99–1.49)	0.056	1.23 (0.97–1.56)	0.085
Vascular severity score (LL)	1.14 (1.02–1.28)	0.027	1.13 (1.00–1.28)	0.051	1.15 (1.01–1.31)	0.030	1.11 (0.97–1.28)	0.131

Model 1: Adjusting for age and sex. Model 2: Adjusting for age, sex, and height. Model 3: Adjusting for age, sex, height, hypertension, DM, dyslipidemia, CVD, CKD, and smoking history. PAD, peripheral arterial disease; OR, odds ratio; CI, confidence interval; CKD, chronic kidney disease; CVD, cardiovascular disease; DM, diabetes mellitus; AMV, abdominal skeletal muscle volume; PMV, psoas muscle volume; TLMV, total leg muscle volume; TMV, thigh muscle volume; LMV, lower leg muscle volume; MD, muscle density; IMATV(ABD), abdominal muscle intermuscular adipose tissue volume; IMATV(TL), total leg intermuscular adipose tissue volume; IMATV(TM), thigh intermuscular adipose tissue volume; IMATV(LL), lower leg intermuscular adipose tissue volume; UL, upper leg; LL, lower leg; TL, total leg.

## Data Availability

The original contributions presented in this study are included in the article. Further inquiries can be directed to the corresponding author.

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
