# Peer review of "The Role of Muscle Density in Predicting the Amputation Risk in Peripheral Arterial Disease: A Tissue Composition Study Using Lower Extremity CT Angiography"

_diagnostics, 2025, doi:10.3390/diagnostics15111439_

Round 1
Reviewer 1 Report
Comments and Suggestions for Authors
Dear Authors,
Thank you for the opportunity to review your manuscript titled "The Role of Muscle Density in Predicting the Amputation Risk in Peripheral Arterial Disease: A Tissue Composition Study Using Lower-limb CT Angiography." I found the study to be highly interesting, well-designed, and clearly written. Only a few minor revisions are needed.
Regarding the Highlights section, I suggest rephrasing the “Problem” to better emphasize the potential opportunity to use automatically evaluated CTA scans as a prognostic tool for clinical risk stratification.
In the Conclusion of the abstract, it would be helpful to underscore that muscle quality may be more critical than quantity, and highlight the value of integrating advanced imaging techniques for risk prediction.
Your Limitations section is well-presented, particularly in addressing the low number of amputation cases. However, I would recommend including a brief comment on whether a power analysis was performed, and if so, whether the number of cases provides sufficient statistical robustness to support your findings.
Overall, I believe this is a strong manuscript that is suitable for publication after minor revisions. Congratulations on this excellent work.
Best regards,
Author Response
Comments 1. Regarding the Highlights section, I suggest rephrasing the “Problem” to better emphasize the potential opportunity to use automatically evaluated CTA scans as a prognostic tool for clinical risk stratification.
Answer: Yes, we thank the reviewer for this valuable comment. We have revised the "Problem" statement in the Highlights section to better emphasize the potential application of automatically evaluated CTA scans in clinical risk stratification, as now reflected on page 1.
2.In the Conclusion of the abstract, it would be helpful to underscore that muscle quality may be more critical than quantity, and highlight the value of integrating advanced imaging techniques for risk prediction.
Answer: Yes, we thank the reviewer for this insightful comment. We have revised the Conclusion in the abstract to emphasize the greater importance of muscle quality over quantity and to highlight the potential role of advanced imaging techniques in clinical risk prediction. This revision is reflected on page 2.
3.Your Limitations section is well-presented, particularly in addressing the low number of amputation cases. However, I would recommend including a brief comment on whether a power analysis was performed, and if so, whether the number of cases provides sufficient statistical robustness to support your findings.
Answer: Yes, we thank the reviewer’s comment on this point. We have addressed this point in detail on page 14 of the manuscript. Specifically, we have added a statement regarding the post-hoc power analysis to clarify the statistical robustness of our findings despite the relatively small number of amputation cases

Reviewer 2 Report
Comments and Suggestions for Authors
The authors attempted to use CT rendering tissue composition to determine the relevance of muscle density as a predictor for amputation. My questions and comments are as followed:
1)as the authors have stated reduced muscle area, increased fatty infiltration, fibrosis, metabolic abnormalities resulted in muscle dysfunction and sarcopenia, the possible causation was not discussed. Its it due to vascular supply compromise or other factors i.e. inflammatory mileau?
2)as these PAD patients, due to leg pain, may not exercise as much as those without PAD, was this taken account into the final analysis.
3)as the study was performed on contrast CT, was vascular collateralization seen? Could it have any impact on MD?
Author Response
The authors attempted to use CT rendering tissue composition to determine the relevance of muscle density as a predictor for amputation. My questions and comments are as followed:
comment 1. as the authors have stated reduced muscle area, increased fatty infiltration, fibrosis, metabolic abnormalities resulted in muscle dysfunction and sarcopenia, the possible causation was not discussed. Its it due to vascular supply compromise or other factors i.e. inflammatory mileau?
Answer: Yes, we thank the reviewer’s comment on this point. We have expanded the discussion on page 14 to address the potential causative mechanisms underlying muscle dysfunction and sarcopenia in PAD patients. While compromised vascular supply is a primary contributor, leading to chronic ischemia and reduced oxygen and nutrient delivery, other factors such as systemic inflammation, oxidative stress, mitochondrial dysfunction, and disrupted anabolic–catabolic signaling also play important roles. These mechanisms are consistent with those observed in age-related sarcopenia, where increased catabolic activity, impaired muscle regeneration, and inflammatory milieu contribute to muscle degradation and fatty infiltration. The causal effects could not be explicitly determined, and at least three potentials” 1 ischemia-induced bio metabolic changes. 2. Ischemia-related physical constraint; 3. Underlined systemic risk factors that cause vasculopathy and muscle metabolic changes. We have address this in Page 14, column 11
Comment 2. as these PAD patients, due to leg pain, may not exercise as much as those without PAD, was this taken account into the final analysis.
Answer: Yes, we thank the reviewer’s comment on this point. The objective physical-activity data (e.g., step counters or formal questionnaires) were not available in our retrospective registry. We have now acknowledged this in the Limitations and clarified why the main conclusions remain valid. We had addressed this in detail in page 22, column 8.
Comment 3. as the study was performed on contrast CT, was vascular collateralization seen? Could it have any impact on MD?
Answer: Yes, we thank the reviewer’s comment on this point. We used non-contrast CT scan, not CTA scan of LECTA for muscle density analysis. Because we aimed to analysis the native MD itself and avoid the mixture effect on MD resulted from perfusion difference in CTA which dependent on patients’ vascularity and contrast injection protocol. Therefore, we did not check the collateral vessels and the muscle perfusion were not take into account. We had addressed this in detail in page 9, column 15.
Round 2
Reviewer 2 Report
Comments and Suggestions for Authors
Issues appropriately addressed